# Immunohistochemical Algorithm for the Classification of Muscle-Invasive Urinary Bladder Carcinoma with Lymph Node Metastasis: An Institutional Study

**DOI:** 10.3390/jcm11247430

**Published:** 2022-12-15

**Authors:** Karla Beatríz Peña, Francesc Riu, Josep Gumà, Francisca Martínez-Madueño, Maria José Miranda, Anna Vidal, Marc Grifoll, Joan Badia, Marta Rodriguez-Balada, David Parada

**Affiliations:** 1Molecular Pathology Unit, Department of Pathology, Hospital Universitari de Sant Joan, 43204 Reus, Spain; 2Institut d’Investigació Sanitària Pere Virgili, 43204 Reus, Spain; 3Facultat de Medicina i Ciències de la Salut, Universitat Rovira i Virgili, 43204 Reus, Spain; 4Institut d’Oncologia de la Catalunya Sud, Hospital Universitari Sant Joan de Reus, IISPV, URV, 43204 Reus, Spain

**Keywords:** bladder, cancer, muscle invasive, metastases, lymph node, immunohistochemical, classification, prognosis, heterogeneity

## Abstract

Muscle-invasive urothelial carcinoma represents 20% of newly diagnosed cases of bladder cancer, and most cases show aggressive biological behavior with a poor prognosis. It is necessary to identify biomarkers that can be used as prognostic and predictive factors in daily clinical practice. In our study, we analyzed different antibodies in selected cases of muscle-invasive urinary bladder carcinoma and lymph node metastasis to identify immunohistochemical types and their value as possible prognostic factors. A total of 38 patients were included, 87% men and 13% women, with a mean age of 67.8 years. The most frequent histopathological type was urothelial carcinoma. In the primary lesion, the mixed type was the most common. In unilateral metastasis, the mixed type was the most frequently found. In cases of primary lesions and bilateral metastasis, the luminal and mixed types were observed. The luminal subtype was the most stable in immunohistochemical expression across primary tumors and metastases. The basal type showed a better prognosis in terms of disease-free survival. In conclusion, immunohistochemical studies are useful in assessing primary and metastatic lesions in patients with urothelial carcinoma. Immunohistochemical classification can typify muscle-invasive urothelial carcinoma, and the immunophenotype seems to have prognostic implications.

## 1. Introduction

Bladder cancer is the tenth most commonly diagnosed cancer worldwide, with ap-proximately 573,000 new cases and 213,000 deaths per year [1,2]. The cancer is four times more frequent in men than in women and is the sixth most common cancer and ninth leading cause of cancer death among men [1,2]. Bladder cancer is a heterogeneous disease associated with diverse clinical outcomes. Thus, tumors that histologically invade the detrusor muscle are called muscle-invasive bladder cancers (MIBCs) and have a greater propensity to spread to lymph nodes and other organs [3,4,5]. When histologic examination shows no invasion into the detrusor muscle, then the cancer is termed nonmuscle invasive and comprises a variety of entities, including carcinoma in situ (CIS), noninvasive papillary tumors, and papillary tumors that invade the lamina propria [3,4,5].

MIBC accounts for approximately 20% of newly diagnosed cases of bladder cancer. Despite radical cystectomy (CR) and pelvic lymph node dissection, approximately 50% of patients will develop disease at distant sites due to disseminated micrometastases [6]. Urothelial carcinoma is the main tumor type, representing approximately 90% of neo-plasms that affect the urinary bladder [2,7]. The epithelium from which urothelial car-cinoma originates is the urothelium, which consists of stratified epithelium throughout the entire urinary system. Several studies have shown that the development of urothelial carcinoma may occur through two pathways, termed papillary and nonpapillary, leading to different but somewhat overlapping subsets of the disease with distinct molecular profiles [7,8,9]. In addition, tumor heterogeneity, both intertumoral and intratumoral, has been widely described [10,11], which makes it difficult to analyze both the primary and metastatic lesions of urothelial carcinoma.

Little is known about the variations in protein expression between primary bladder cancer and lymph node metastases. The aim of the present study was to compare the immunohistochemical (IHC) findings of patients with muscle-invasive urothelial carcinoma (MIUC) and lymph node metastases. In addition, the possibility of applying an IHC algorithm that allows for the classification of MIUC was evaluated. Finally, the prognostic value of the classification in patients with MIUC was investigated.

## 2. Materials and Methods

### 2.1. Patients

This retrospective and descriptive cohort study was performed in the MIUC of the urinary bladder patients who had undergone radical cystectomy and bilateral ileo-obturator lymphadenectomy. The study protocol was reviewed and approved by the Ethics Committee of the Sant Joan University Hospital in Reus (registration number CEIC11–04-28/4PROJ3), and written informed consent was obtained from each subject in accordance with the 1964 Helsinki Declaration and its subsequent amendments.

Thirty-eight patients with muscle invasive urothelial carcinoma who had undergone radical cystectomy and pelvic lymph node dissection were included from the Sant Joan University Hospital in Reus. The general criteria for patient selection were as follows: treated from January 2014 to January 2019; biopsy with a confirmed diagnosis of MIUC; uni- and/or bilateral lymph node metastases; absence of severe psychiatric disorders, chronic alcoholism or drug addiction; and adequate understanding of the surgery and adherence to follow-up standards. Patient charts were reviewed to collect data regarding sex, age at diagnosis, clinical stage, surgery, residual disease after surgery, systemic treatment, local recurrence, and survival.

### 2.2. Histopathological Study

The cystectomy specimens were opened through the urinary bladder and fixed in 10% buffered formalin for at least 48 h. After fixation, the biopsied sections from macroscopic lesions were embedded in paraffin, cut into 2 µm sections, and stained with hematoxylin and eosin (H&E). All dissected lymph node material was embedded in paraffin and prepared following the previous staining protocol. For each case of radical cystectomy, tumor histology, grade (according to the 2022 World Health Organization) [12], pathological stage, presence of carcinoma in situ (CIS), lymphovascular invasion, and margin status were analyzed. In cases of lymph node dissection, the total number of lymph nodes, number of metastatic lymph nodes, size of large metastases, and extra-capsular lymph node involvement were evaluated.

### 2.3. Immunohistochemical Study

Two-micrometer sections were obtained from both paraffin-embedded MIUC and lymph node metastases samples and were placed in a VENTANA^®^ Benchmark UL-TRA/LT immunohistochemistry automatic processor, Ventana Medical Systems, USA, using the standardized protocol for uroplakin, GATA3, cytokeratin 5, cytokeratin 14, cytokeratin 18, cytokeratin 20, and CD44, including retrieval solution, pH 9, and a detection kit for Immunohistochemistry Optiview^®^ DAB (VENTANA^®^). The primary anti-uroplakin, GATA3, cytokeratin 5, cytokeratin 14, cytokeratin 18, cytokeratin 20, and CD44 antibodies (prediluted) (Phoenix Pharmaceutical, Inc.) were incubated for 32 min. Finally, the IHC sections were revealed with diaminobenzidine, contrasted with Meyer’s hematoxylin, and examined under an Olympus BX41 light microscope, with direct in-creases in magnification ranging from 2× to 60×. The IHC images of both MIUC and lymph node metastases were evaluated by two independent pathologists.

### 2.4. Semiquantitative Evaluation of Immunoreactivity

IHC interpretation was performed using combined intensity and percentage scales defined by different groups to assess IHC profiles [8,13,14,15]. All tumor areas of both MIUC and lymph node metastases were evaluated. Positive IHC staining for an antibody was considered when cytoplasmic or nuclear staining was observed, depending on the antibody tested. Each antibody was assigned a percentage of positivity, with the following positivity intervals: 0% as negative, between 1 to 10%, between 11 to 50%, between 51 to 80%, and greater than 80%. Additionally, intense cytoplasmatic IHC staining was scored as follows: negative: no staining; 1+: weak staining; 2+: moderate staining, and 3+: intense staining. Appropriate positive and negative controls were used for each antibody. Additionally, the reactivity of the positive controls served to assess the intensity of the staining. The magnification varied between 4×, 10× and 20×.

### 2.5. Statistical Analysis

Descriptive statistics are presented as N (%) for qualitative clinical variables, while median, 25th percentile (P25) and 75th percentile (P75) are used for quantitative clinical variables. Statistical analyses were carried out in R (version 4.2.0).

Hierarchical clustering was performed on the immunohistochemistry (IHC) antibody expression in bladder tissue using the pvclust R package (version 2.2-0) with default settings, which allowed us to assess the robustness of each cluster by bootstrapping with resampling (nboot = 1000). The clusters with an approximate unbiased (AU) *p* value ≥ 95 were deemed statistically significant (significance level < 0.05) and used in survival analysis, while samples not robustly assigned to any cluster (unassigned) were excluded from survival analysis. The clusters were then labeled according to CK20 and CK18 bladder tissue expression as luminal, basal or mixed type and according to CK5 and GATA bladder tissue expression as CK5± or GATA± subtypes. Heatmaps of IHC antibody expression were generated with the ComplexHeatmap R package (version 2.12.0) using the dendrogram from bootstrap hierarchical clustering (top dendrogram).

Survival analysis was performed with the survival (version 3.3-1) and survminer R packages (version 0.4.9). Progression-free survival (PFS) is defined as the time from cystectomy surgery until a detected progression according to the Response Evaluation Criteria in Solid Tumors (RECIST) or date of last contact (right-censored point). Overall survival was defined as the time from cystectomy surgery until a known death event or date of last contact (right-censored point).

The suggested algorithm for MIBC subtype classification included a 20% expression cutoff. Alternative cutoffs are also represented for each cluster.

## 3. Results

### 3.1. Clinical Findings

A total of 38 patients were included in the study, comprising 33 (87%) men and 5 (13%) women, with a mean age of 67.8 years (62–75 years). The most frequent pathological stage (pT) was T3 in 18 patients (50%), followed by T4 in 15 patients (40%) and finally stage T2 in 4 patients (10%). Cystectomy was the only initial treatment in 30 patients (79%), in the other 8 of the 38 included patients received a combined treatment of cystectomy and chemotherapy (CT) (4 patients, 11%), cystectomy plus immunotherapy (IT) (2 patients, 5%), combination of cystectomy plus chemotherapy and immunotherapy (1 patient, 3%) and surgical treatment plus radiotherapy (RT) (1 patient, 3%). A total of 18 patients (47%) showed progression of urothelial carcinoma, and 24 patients (63%) died of bladder carcinoma. Table 1 summarizes the clinical findings.

### 3.2. Histopathological Findings

Urothelial carcinoma was the most frequent histological type, found in 14 tumors (37%), followed by urothelial carcinoma with divergent squamous differentiation in 7 tumors (18%). Five tumors (13%) were pure squamous cell carcinomas, and 12 carcinomas (32%) showed combinations of different histological types of carcinomas, such as solid urothelial carcinoma with micropapillary carcinoma, sarcomatoid urothelial carcinoma, urothelial carcinoma with glycogen-rich carcinoma, and urothelial carcinoma with plasmacytoid carcinoma (Figure 1). The histopathological study of the pelvic lymph node dissection samples showed that between 1 and 25 lymph nodes (mean: 7.86) were evaluated in right lymphadenectomy and between 1 and 20 lymph nodes (mean: 8.53) were evaluated in left lymphadenectomy. In total, 23 (60.53%) patients showed unilateral lymph node metastasis, 11 patients out of 23 showed lymph node metastasis on only the right side (1–3 metastatic nodes, mean: 1.64), and in the remaining 12 patients, only the left side was affected (1–3 metastatic nodes, mean: 1.92). In 15 patients (39.47%), bilateral lymph node metastasis was observed (1–17 metastatic lymph nodes, mean: 4.24).

### 3.3. Immunohistochemical Findings

#### 3.3.1. Muscle-Invasive Urothelial Carcinoma

The IHC study showed that GATA3 was the most frequently expressed marker, in 94.74% of the patients, followed by cytokeratin 18 (92.11%) and cytokeratin 5 (89.47%). Uroplakin was expressed in 26.32% of the patients. The immunoreaction intensity ranged from 2+ to 3+, and the percentage of positive cells ranged from 1% to 90%. In general, the expression of basal markers (CK5, CK14 and CD44) was seen in 68.51% of the patients, while luminal markers (uroplakin, CK20, and CK18) were demonstrated in 55.27% of the patients (Figure 2).

To try to define specific clusters that would facilitate defining specific subtypes of urothelial carcinoma, the bootstrap clustering algorithm (*n* = 1000) was applied and showed clusters with significant differences between groups (*p* < 0.05). Subsequently, heatmap analysis was able to identify basal and luminal groups based on the expression of luminal markers, such as cytokeratin 20 and cytokeratin 18 or on the expression of basal markers, such as cytokeratin 5 and 14. Cluster 5 showed ex-pression of luminal markers and basal markers (cytokeratin 18 and cytokeratin 5) and was considered a mixed immunophenotype. Finally, cluster 6 showed a loss of cytokeratin 20 despite expressing cytokeratin 10 (Figure 3).

Based on the results obtained in bootstrap clustering and heatmap analysis, the following algorithm was proposed for the classification of MIUCs (Figure 4), with a cutoff of 20% for each antibody analyzed (Figure 4 and Figure 5):

By applying the algorithm based on two markers for MIUCs, it was possible to classify 11 patients (28.95%) as having a luminal immunophenotype, 10 patients (26.32%) as having a basal immunophenotype, and 16 patients (42.11%) as having a mixed immunophenotype. One patient (2.63%) could not be classified with a morphology that corresponded to urothelial carcinoma with divergent squamous differentiation and a plasmacytoid carcinoma component.

#### 3.3.2. Unilateral Pelvic Lymph Node Metastasis

In general, the immunohistochemical study showed that GATA3 was the most frequently expressed marker in 86.96% of metastatic pelvic lymph nodes, followed by cytokeratin 5 (91.30%), and cytokeratin 18 (88.96%). Uroplakin was evidenced in 20.09%. The immunoreaction intensity was from 2+ to 3+, and the percentage of positive cells varied between 1% to 90. The expression of basal markers (CK5, CK14 and CD44) was evidenced in 62.32% of the patients, while luminal markers (uroplakin, CK20, and CK18) were demonstrated in 54.96% of cases. Applying the algorithm of 2 markers pelvic lymph node metastatic carcinomas could be classified in 14 patients (36.84%) as luminal immunophenotype, 8 patients (21.05%) as basal immunophenotype, and 16 patients (42.11%) were classified as mixed immunophenotype.

#### 3.3.3. Bilateral Pelvic Lymph Node Metastasis

In patients with bilateral pelvic lymph node metastases, GATA3 could be observed in 93.33% and 100% of the two regions analyzed, followed by cytokeratin 18 (100% of the two regions) and cytokeratin 5 (86.67% of the two regions). Cytokeratin 20 was detected in 66.67% and 80% of patients, cytokeratin 14 in 60% and 33.33%, CD44 in 33.33% and 26.67%, and uroplakin in 26.67% and 40%. When the cases were analyzed in pairs (*n* = 15), the expression of basal markers (CK5, CK14 and CD44) was observed in 60% and 48.89% of each region studied, while the expression of luminal markers (uroplakin, CK20, and CK18) was demonstrated in 64.45% and 73.33% of the cases according to each region analyzed. Eight lymph node metastases (53.33%) were classified as having a luminal immunophenotype, and the remaining seven lymph node metastases (46.67%) were classified as having a mixed immunophenotype. 

#### 3.3.4. Urothelial Carcinoma and Unilateral Pelvic Lymph Node Metastasis

Using the immunophenotype classification algorithm, it was possible to show that in the 11 patients with a luminal immunophenotype, 10 maintained this immunophenotype in the pelvic lymph node metastases (90.91%), and the remaining patient (9.09%) showed a basal immunophenotype in the pelvic lymph node metastasis. Overall, the baseline primary immunophenotype (10 patients) remained the same immunophenotype in 6 patients (60%), while the immunophenotype became mixed in 3 patients (30%) and luminal in 1 patient (10%). The mixed primary immunophenotype (16) was seen in the lymph node metastases of 12 patients (75%); 3 metastatic carcinomas (18.75%) were classified as luminal; and 1 carcinoma (6.25%) was classified as basal. The patient whose primary bladder lesion could not be classified showed a mixed immunophenotype in the pelvic lymph node metastasis.

#### 3.3.5. Urothelial Carcinoma and Bilateral Pelvic Lymph Node Metastasis (Paired Cases)

In the 15 patients with bilateral pelvic lymph node metastasis, the immunophenotype classification algorithm was able to show that in the 6 patients with luminal immunophenotype 5 and 6, the immunophenotype maintained in the pelvic lymph node metastases. Overall, the primary basal immunophenotype observed in the two primary lesions were basal in one lesion and mixed in one of the metastases, and on the contralateral side, the immunophenotype was mixed in both. The primary mixed immunophenotype (6) was evidenced in the pelvic lymph node metastases of 4 patients (75%), and the remaining two metastases were luminal; among the contralateral metastases, five were classified as mixed, and the remaining 3 were classified as luminal.

Kaplan–Meier analysis for PFS was performed in the immunohistochemistry (IHC) groups. The IHC groups suggested a significant association between group identity and the PFS rate. Specifically, the luminal and basal clusters from the identified seven-marker expression cluster were significantly associated with PFS rate. Kaplan-Meier analysis for PFS was also performed to assess the simplified two-marker algorithm, which could nearly replicate the resulting bootstrap clusters and also showed a significant association with PFS rate (Figure 6a,b). No significant differences were observed between unilateral and bilateral pelvic lymph node metastases (Figure 6 and Figure 7).

## 4. Discussion

Invasive urothelial carcinoma is the most common malignant neoplasm of the urinary tract. MIUCs represent approximately 20–25% of all cancers in the urinary bladder [16], and most of them show rapid progression and metastasis and are associated with a poor prognosis. In the present study, we analyzed a series of patients who underwent radical cystectomy for a diagnosis of MIUC, showing cancer progression in 47% of patients and a cancer-specific mortality rate of 63%. In addition, all patients evaluated had lymph node metastases, which may negatively affect the outcome [17]. Our findings support that MIUC is an aggressive neoplasm with a poor prognosis, and it is necessary to identify new markers that can be used as prognostic and predictive factors in this cancer.

Bladder cancer is often histologically heterogeneous within a given patient. This is well demonstrated by the frequent coexistence of conventional urothelial carcinoma and named histologic variants of bladder cancer, such as squamous and micropapillary cancers [11]. In the present study, the histopathological heterogeneity in MIBC was con-firmed, since approximately 50% of the cancers evaluated were urothelial carcinoma with a divergent squamous differentiation and a combination of morphological variants, including more aggressive variants, such as micropapillary carcinoma, sarcomatoid carcinoma, and plasmacytoid carcinoma [18,19]. An interesting finding was that five patients who were diagnosed after extensive pathologic examination of the carcinoma had pure squamous cell carcinoma, without evidence of a urothelial carcinoma component. This finding represents a three times higher rate of pure squamous cell carcinoma than the reported incidence and could be related to the possible etiological factors associated with this cancer, such as smoking, since these patients had a higher intensity of smoking [12]. The recent classification of tumors of the genitourinary system of 2022 noted the importance of reporting the different components of conventional urothelial carcinoma, as well as the histological subtypes and the divergent differentiation in each tumor. The need to quantify these elements was also indicated, given their possible implications for patient management [19].

Tumor heterogeneity in MIUC, in addition to morphological variability, also includes diversity from a molecular point of view [19,20]. This molecular diversity implies different omic expressions [21], such as at the protein level. The study of protein expression has made it possible to discover different cellular components in the urothelium, which has facilitated the characterization of umbrella cells, superficial cells, intermediate cells and basal cells [22]. In addition, this knowledge has made it possible to establish the dual-track concept of bladder carcinogenesis [7]. This concept suggests that parabasal cells give rise to superficial papillary lesions through the papillary luminal pathway, while basal cells that progress through the nonpapillary basal pathway give rise to invasive lesions [7]. In the present study, we demonstrated variable expression of both luminal and basal markers in MIUC, with a higher expression of basal markers. This finding differs from that reported by other studies in which MIUC was analyzed by whole-transcriptome mRNA expression profiling, and the luminal type was found to be the most frequent, followed by the basal and double negative subtypes [23]. However, from an IHC point of view, our findings confirm the heterogeneity of protein expression in this type of cancer and that invasive lesions, in general, express markers related to the nonpapillary basal pathway.

In the present study, to develop a potential IHC classifier of molecular subtypes of MIBC and lymph node metastasis that can be used in routine clinical practice, we analyzed 7 previously described markers [7,13,14,15]. Our analysis showed that cytokeratin 20 and cytokeratin 18 are the two markers that enabled differentiation between the luminal and basal subtypes. In addition, our study confirmed the usefulness of the 20% cutoff point to interpret the positivity of the investigated markers, showing coincident results with those published by other studies on the expression of basal and luminal markers in urothelial carcinoma [8,15]. In addition, the classification of subtypes in MIUC in our study showed significant differences in overall survival between the groups analyzed, with a worse prognosis for the luminal subtype than for the basal and mixed subtypes. This finding differs from what has been shown in other studies in which the basal subtype was associated with a worse prognosis [24]; however, other researchers have not shown differences in DSS or other clinical prognostic factors suggesting a better prognosis for the basal subtype, even though the results showed trends without statistical significance [25]. A possible explanation for this difference is that we included patients with advanced stages and lymph node metastasis, which requires therapeutic interventions. This may imply that basal tumors respond better to these therapies. Another factor that could affect this difference is the expression of genes related to tumor metabolism, which would imply differences in immunotherapy response, as well as variable responses to cisplatin, doxorubicin, and other first-line anticancer drugs [26].

Another aspect of our study consisted of studying the evolution of the expression of IHC markers, comparing bladder tumors and lymph node metastases. The relationship between the primary tumor and unilateral lymph node metastasis showed that the luminal immunophenotype was the most stable, since in bladder carcinomas with this immunophenotype, 90% showed the same luminal type in the unilateral lymph node metastases, and only one case showed a basal immunophenotype. On the other hand, the basal type maintained its immunophenotype in 60% of the unilateral lymph node metastases and changed to luminal and mixed immunophenotypes in the rest of the patients. Finally, the mixed immunophenotype persisted in 75% of the patients, showing basal and luminal immunophenotypes in the rest of the patients. Sjödahl et al. [27] studied MIUC and lymph node metastases using immunohistochemistry and gene expression profiling, and they showed that the basal/squamous-like subtype was the most discordant type. Our results confirm that the basal and mixed immunophenotypes are the most discordant in terms of immunohistochemistry and demonstrate the IHC plasticity of both immunophenotypes, while the luminal type was more stable in terms of its immunophenotypic expression in patients with unilateral lymph node metastasis. In patients with bilateral lymph node metastases (paired cases), the immunophenotypic variations were greater in one of the two metastatic regions evaluated. In general, the basal immunophenotype was not found in one of the metastatic regions, while the luminal and mixed immunophenotypes persisted in lymph node metastases. Further studies are necessary to try to understand the significance of the immunophenotypic variations among MIUC, unilateral and/or bilateral metastases and the prognostic and predictive role of these variations in these patients.

Our study has some limitations, such as the low number of patients included in the study, although our sample comprises a particular group of patients with MIUC and lymph node metastasis. Additionally, the imaging study was carried out semi-quantitatively; however, this analysis represents a useful tool in daily clinical practice and can serve as a basis for using specialized tools for quantitative image analysis.

## 5. Conclusions

In summary, our study demonstrated that MIUC represents a heterogeneous disease at different stages, with fundamental clinical implications for its prognosis and serves as the basis for possible predictive factors. The IHC study was a useful tool to assess primary and metastatic lesions in patients with urothelial carcinoma, and by applying a classification system based on two markers, the different subtypes of MIUC could be typified. The classification of the different subtypes seems to have prognostic implications and could help to stratify patients.

## Figures and Tables

**Figure 1 jcm-11-07430-f001:**
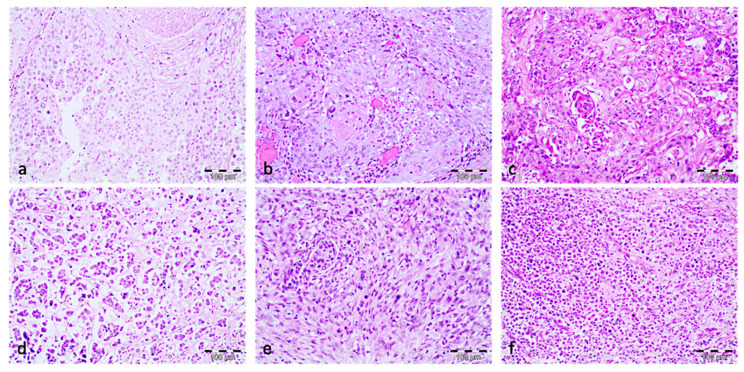
Histopathological findings of muscle-invasive urothelial cell carcinoma. (**a**) Urothelial cell carcinoma. (**b**) Urothelial cell carcinoma with divergent squamous differentiation. (**c**) Pure squamous cell carcinoma. (**d**) Micropapillary carcinoma. (**e**) Sarcomatoid carcinoma. (**f**) Plasmacytoid carcinoma. (HE, 10×).

**Figure 2 jcm-11-07430-f002:**
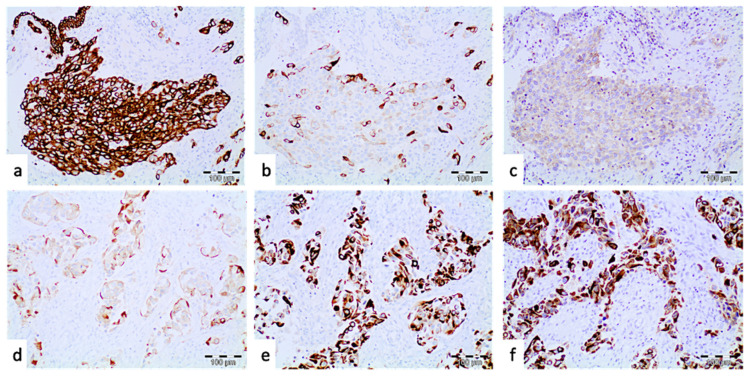
Inmunohistological findings in muscle-invasive urothelial cell carcinoma basal markers (**a**–**c**). (**a**) Diffuse and strong expression of cytokeratin is shown. (**b**) Cytokeratin 14 showing focal cytoplasmic positivity. (**c**) CD44 cytoplasmic expression in neoplastic urothelial cells. Luminal markers (**d**–**f**). (**d**) Uroplaquin showing cytoplasmic positivity in neoplastic cells. (**e**) Neoplastic cells with cytoplasmic cytokeratin 20 expression. (**f**) Diffuse and intensive cytoplasmic expression of cytokeratin 18 is shown. (Diaminobenzidine (DAB), 10×).

**Figure 3 jcm-11-07430-f003:**
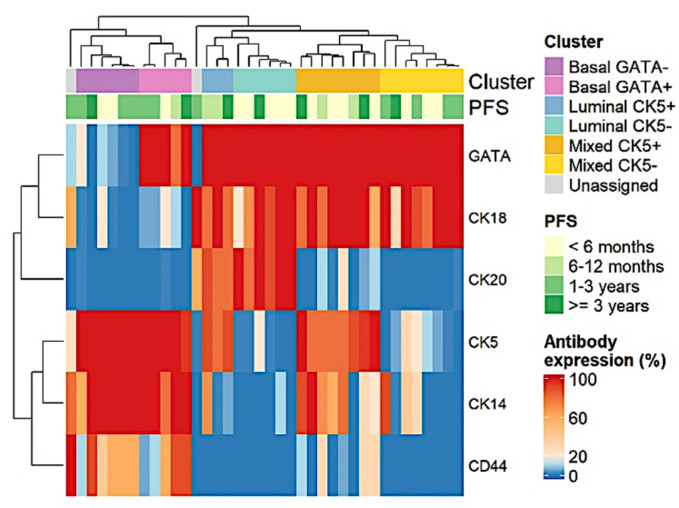
Heatmap of antibody expression (%). Hierarchical clustering with bootstrapping identified 6 clusters, which were labeled luminal, basal or mixed according to the expression of the markers CK20 and CK18. Luminal clusters expressed both markers, mixed clusters expressed only CK18, while basal clusters lacked both markers. Clusters were then further subdivided and labeled according to the expression of CK5 or GATA.

**Figure 4 jcm-11-07430-f004:**
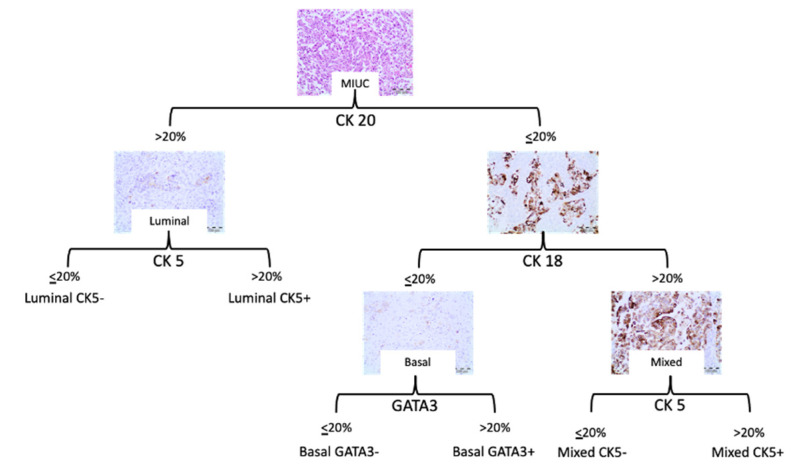
Proposed algorithm based on two markers and the representation of the bootstrap clustering results and heatmap analysis.

**Figure 5 jcm-11-07430-f005:**
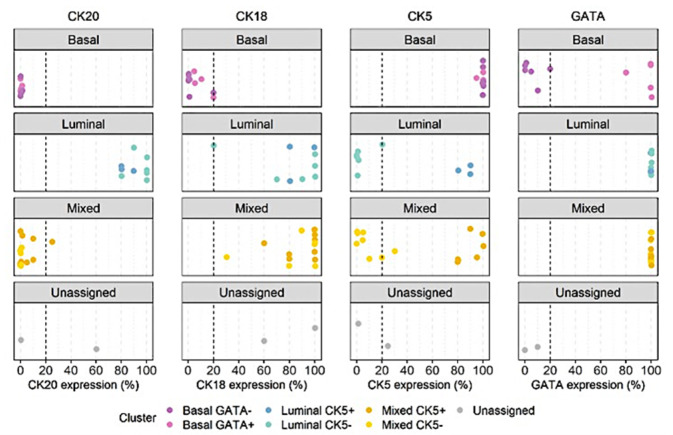
Antibody expression showing the 20% cutoff (dotted vertical line) for each cluster.

**Figure 6 jcm-11-07430-f006:**
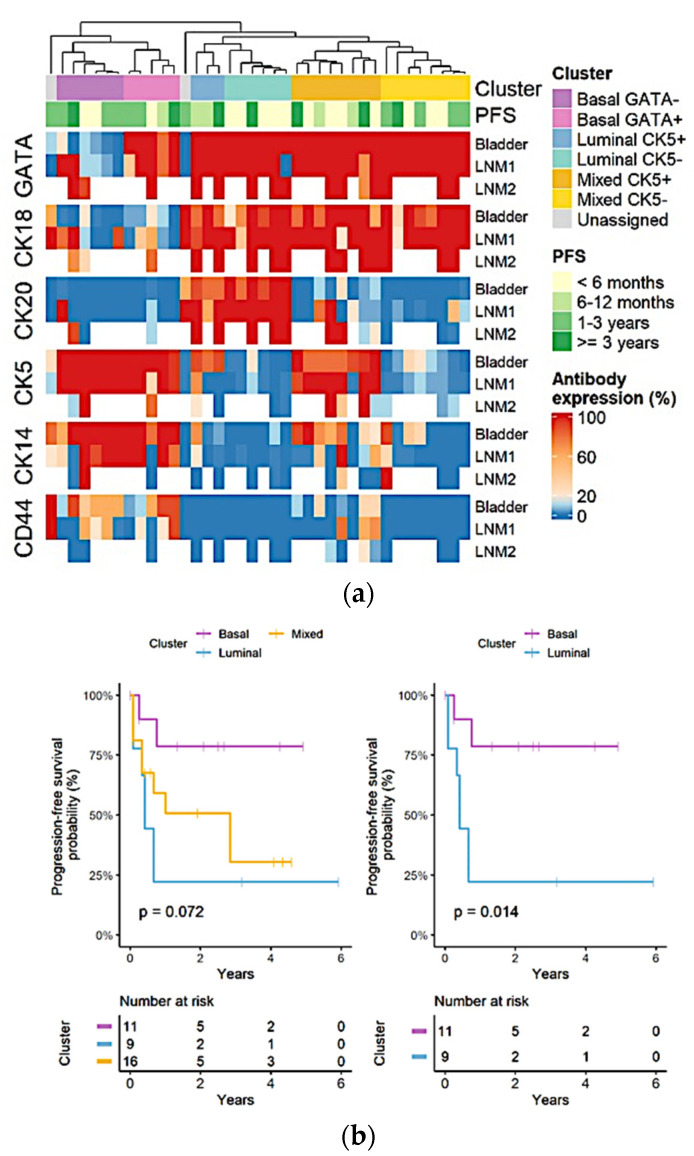
Heatmap of antibody expression (%) for bladder and pelvic lymph node metastasis tissues (**a**). Survival analysis of IHC clusters suggested a significant association between cluster identity and progression-free survival (PFS) rate. Specifically, luminal and basal IHC clusters were significantly associated with PFS rates (**b**).

**Figure 7 jcm-11-07430-f007:**
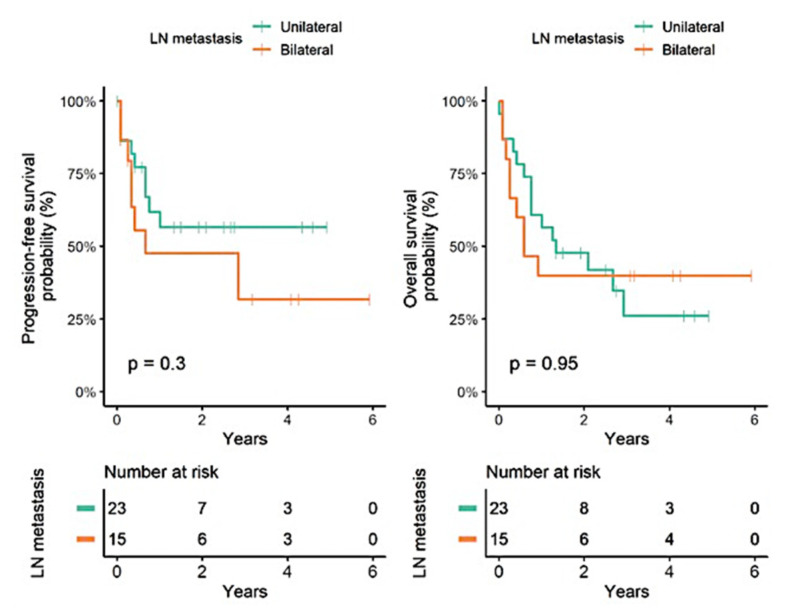
Survival analysis for unilateral or bilateral pelvic lymph node metastasis.

**Table 1 jcm-11-07430-t001:** Clinical findings in patients with muscle-invasive urinary bladder carcinoma and lymph node metastases (*n* = 38).

	Urothelial	Urothelial Combined	Others	All
	*n* = 14	*n* = 14	*n* = 10	*n* = 38
Age				
Mean	68.4 years	68.2 years	66.4 years	67.8 years
Median (P25, P75)	66.5 (63, 77)	69.5 (63, 73)	65.5 (59, 73)	67.5 (62, 75)
Sex				
Men	13 (93%)	12 (86%)	8 (80%)	33 (87%)
Women	1 (7%)	2 (14%)	2 (20%)	5 (13%)
Stage				
pT2	1 (7%)	2 (14%)	1 (10%)	4 (10%)
pT3	5 (36%)	7 (50%)	7 (70%)	18 (50%)
pT4	8 (57%)	5 (36%)	2 (20%)	15 (40%)
Unilateral or bilateral pelvic lymph node metastases				
Unilateral (NT1)	10 (71%)	9 (64%)	4 (40%)	23 (61%)
Bilateral (NT2)	4 (29%)	5 (36%)	6 (60%)	15 (39%)
Distant metastases				
M0	9 (64%)	14 (100%)	9 (90%)	32 (84%)
M1	5 (36%)	0	1 (10%)	6 (16%)
Morphology				
Pure urothelial	14 (100%)	0	0	14 (37%)
Urothelial + squamous	0	7 (50%)	0	7 (18%)
Pure squamous	0	0	5 (50%)	5 (13%)
Others	0	7 (50%)	5 (50%)	12 (32%)
Primary treatment				
Only cystectomy	10 (71%)	11 (79%)	9 (90%)	30 (79%)
Cystectomy + CT	2 (14%)	1 (7%)	1 (10%)	4 (11%)
Cystectomy + IT	1 (7%)	1 (7%)	0	2 (5%)
Cystectomy + CT + IT	1 (7%)	0	0	1 (3%)
Cystectomy + RT	0	1 (7%)	0	1 (3%)
Adjuvant chemotherapy				
N (%)	5 (36%)	11 (79%)	4 (40%)	20 (53%)
Progression				
N (%)	5 (36%)	6 (43%)	2 (20%)	13 (34%)
Death				
N (%)	9 (64%)	8 (57%)	6 (60%)	23 (61%)

Abbreviations: CT = chemotherapy; IT = immunotherapy; RT = radiotherapy; SD = standard deviation; Q25 = quartile 25; Q75 = quartile 75.

## Data Availability

All of the data are present in the manuscript.

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
