# Peer review of "Immunohistochemical Algorithm for the Classification of Muscle-Invasive Urinary Bladder Carcinoma with Lymph Node Metastasis: An Institutional Study"

_jcm, 2022, doi:10.3390/jcm11247430_

Round 1

Reviewer 1 Report

The authors presented an interesting paper on substaging of MIBC, yet in a relatively low-numbered cohort. Moreover, the group was highly heterogenous considering e.g. the treatment (i.e. chemotherapy/immunotherapy in unknown setting), number of nodes sampled or pathological findings (e.g. histopathological type). Although the authors commented on the heterogeneity of the disease, one can hardly judge about the final results due to rather descriptive than comparative analysis. Among other minor comments it is worth mentioning that the ilio-obturator term is rather confusing, as the guidelines strongly suggest to perform extended pelvic Lnd.

Author Response

Reviewer 1:

The authors presented an interesting article on the substaging of MIBC, albeit in a relatively small cohort. Furthermore, the group was very heterogeneous considering e.g. treatment (ie, chemotherapy/immunotherapy in an unknown setting), number of nodes sampled, or pathologic findings (eg, histopathologic type). Although the authors commented on the heterogeneity of the disease, one can hardly judge the final results due to a descriptive rather than comparative analysis. Among other minor comments, it is worth mentioning that the term ilio-obturator is quite confusing, as the guidelines strongly suggest performing extended pelvic Lnd.

Dear reviewer, thank you for your valuable comment and time. We have collected your comments in order to improve our work and so that it can be assessed for possible publication.

1) Point 1: The authors presented an interesting article on the substaging of MIBC, albeit in a relatively small cohort.

1.1) Response to point 1: As you point out, we present an article on MIBC, in addition to the presence of regional nodal metastases. The presence of pelvic lymph node metastasis at the time of radical cystectomy represents approximately 25% of MIUC that undergo cystectomy (PMID: 30104615  DOI: 10.1038/s41585-018-0066-1). Since our study focused on patients with MIUC with lymph node metastasis, this could explain the sample size included in our work.

2) Point 2: The group was very heterogeneous considering e.g. treatment (ie, chemotherapy/immunotherapy in an unknown setting), number of nodes sampled, or pathologic findings (eg, histopathologic type).

2.1) Response to point 2: We fully agree with your assessment, since MIUC represents a challenge due to its morphological, molecular and therapeutic heterogeneity. In our approach we used a diagnostic algorithm, with seven immunohistochemical markers, which could be used in daily clinical practice for the classification of MIUC, demonstrating the heterogeneity at the proteomic level in this neoplasm.

3) Point 3: Although the authors commented on the heterogeneity of the disease, one can hardly judge the final results due to a descriptive rather than comparative analysis.

3.1) Response to point 3: Thank you for your valuable comment, and in the discussion of our study, we compare our results with those reported by other authors at different levels, such as:

- At the histopathological level:

Line 307-320: In the present study, the histopathological heterogeneity in MIBC was con-firmed, since approximately 50% of the cancers evaluated were urothelial carcinoma with a divergent squamous differentiation and a combination of morphological variants, including more aggressive variants such as micropapillary carcinoma, sarcomatoid carcinoma, and plasmacytoid carcinoma [18,19]. An interesting finding was that 5 patients who were diagnosed after extensive pathologic examination of the carcinoma had pure squamous cell carcinoma, without evidence of a urothelial carcinoma component. This finding represents a three times higher rate of pure squamous cell carcinoma than the reported incidence and could be related to the possible etiological factors associated with this cancer, such as smoking, since these patients had a higher intensity of smoking [12]. The recent classification of tumors of the genitourinary system of 2022 noted the importance of re-porting the different components of conventional urothelial carcinoma, as well as the histological subtypes and the divergent differentiation in each tumor. The need to quantify these elements was also indicated, given their possible implications for patient management [19].

- At the omic level:

Line 338-357: In the present study, to develop a potential IHC classifier of molecular subtypes of MIBC and lymph node metastasis that can be used in routine clinical practice, we ana-lyzed 7 previously described markers [7,13-15]. Our analysis showed that cytokeratin 20 and cytokeratin 18 are the two markers that enabled differentiation between the luminal and basal subtypes. In addition, our study confirmed the usefulness of the 20% cutoff point to interpret the positivity of the investigated markers, showing coincident results with those published by other studies on the expression of basal and luminal markers in urothelial carcinoma [8,15]. In addition, the classification of subtypes in MIUC in our study showed significant differences in overall survival between the groups analyzed, with a worse prognosis for the luminal subtype than for the basal and mixed subtypes. This finding differs from what has been shown in other studies, in which the basal subtype was associated with a worse prognosis [24]; however, other researchers have not shown dif-ferences in DSS or other clinical prognostic factors suggesting a better prognosis for the basal subtype, even though the results showed trends without statistical significance [25]. A possible explanation for this difference is that we included patients with advanced stages and lymph node metastasis, which requires therapeutic interventions. This may imply that basal tumors respond better to these therapies. Another factor that could affect this difference is the expression of genes related to tumor metabolism, which would imply differences in immunotherapy response, as well as variable responses to cisplatin, dox-orubicin, and other first-line anticancer drugs [26].

4) Point 4: Among other minor comments, it is worth mentioning that the term ilio-obturator is quite confusing, as the guidelines strongly suggest performing extended pelvic Lnd.

4.1) Response to point 3: Thank you for your valuable comment, and we have changed the term ilio-obturator to pelvic lymph node dissection.

Reviewer 2 Report

In this work, the authors analyzed different antibodies in selected muscle-invasive urinary bladder carcinoma and lymph node metastasis to identify immunohistochemical types and their value as possible prognostic factors. The authors used samples from thirty-eight patients for histological analysis. I have gone through the manuscript, and I found the topic and the work done of great interest, and suitable for publication in the “Journal of Clinical Medicine”. The work presented is diversified and includes many important results. I recommended the manuscript for publication in the “Journal of Clinical Medicine” after considering the following major points:

1-    The sample size is a moderately small number and mostly 67 years

2-    The authors depended on histological examination as a prognostic and predictive marker, but this method is invasive and I think measuring blood and urine markers will be less invasive and faster

3-    Studying the tumor stage is unclear

Author Response

In this work, the authors analyzed different antibodies in selected muscle-invasive urinary bladder carcinoma and lymph node metastasis to identify immunohistochemical types and their value as possible prognostic factors. The authors used samples from thirty-eight patients for histological analysis. I have gone through the manuscript, and I found the topic and the work done of great interest, and suitable for publication in the “Journal of Clinical Medicine”. The work presented is diversified and includes many important results. I recommended the manuscript for publication in the “Journal of Clinical Medicine” after considering the following major points:

1-    The sample size is a moderately small number and mostly 67 years.

2-   The authors depended on histological examination as a prognostic and predictive marker, but this method is invasive and I think measuring blood and urine markers will be less invasive and faster.

3-    Studying the tumor stage is unclear.

Dear reviewer, thank you for your valuable comment and time. We have collected your comments in order to improve our work and so that it can be assessed for possible publication. Here are the changes made:

1) Point 1: The sample size is a moderately small number and mostly 67 years

1.1) Response to point 1: As you point out, we present an article on MIBC, in addition to the presence of regional nodal metastases. The presence of pelvic lymph node metastasis at the time of radical cystectomy represents approximately 25% of MIUC that undergo cystectomy (PMID: 30104615  DOI: 10.1038/s41585-018-0066-1). Since our study focused on patients with MIUC with lymph node metastasis, this could explain the sample size included in our work.

2) Point 2: The authors depended on histological examination as a prognostic and predictive marker, but this method is invasive and I think measuring blood and urine markers will be less invasive and faster.

2.1) Response to point 2: Thank you very much for your comment and We agree with your point of view on the need to know new markers that allow us to quickly know prognostic and predictive factors. In this sense, we use epigenetics in urine samples for the follow-up of non-MIUC patients (PMID: 35807141  PMCID: PMC9267544  DOI: 10.3390/jcm11133855), which reduces the application of invasive procedures, benefiting the patient and the health system. However, identifying markers that can be used in daily clinical practice is a challenge, and in the present study we propose an algorithm that allows the molecular classification of MIUC, for the purposes of standardization and accessibility.

3) Point 3: Studying the tumor stage is unclear.

3.1) Response to point 3: Thank you for your valuable feedback, and we use the current AJCC 8 edition staging system: Amin MB, Edge SB, Greene FL, et al. AJCC Cancer Staging Manual. ed. 8 Cham, Switzerland: Springer; 2017

Round 2

Reviewer 1 Report

Thank you for your comments and thorough corrections.